# Clinical Impact of New Treatment Strategies for HER2-Positive Metastatic Breast Cancer Patients with Resistance to Classical Anti-HER Therapies

**DOI:** 10.3390/cancers15184522

**Published:** 2023-09-12

**Authors:** Marta Tapia, Cristina Hernando, María Teresa Martínez, Octavio Burgués, Cristina Tebar-Sánchez, Ana Lameirinhas, Anna Ágreda-Roca, Sandra Torres-Ruiz, Iris Garrido-Cano, Ana Lluch, Begoña Bermejo, Pilar Eroles

**Affiliations:** 1Department of Clinical Oncology, University Clinical Hospital of Valencia, 46010 Valencia, Spain; martapis3@gmail.com (M.T.); c.hernandomelia@gmail.com (C.H.); maitemartinez3@yahoo.es (M.T.M.); ctebarincliva@gmail.com (C.T.-S.); lluch_ana@gva.es (A.L.); begobermejo@gmail.com (B.B.); 2Biomedical Research Institute INCLIVA, 46010 Valencia, Spain; analameirinhas@gmail.com (A.L.); annaagreda@gmail.com (A.Á.-R.); sand.torres95@gmail.com (S.T.-R.); iris_gc_255@hotmail.com (I.G.-C.); 3Department of Pathology, Hospital Clinic of Valencia, 46010 Valencia, Spain; octavio.burgues@uv.es; 4Biomedical Research Networking Center in Oncology (CIBERONC), 28029 Madrid, Spain; 5Interuniversity Research Institute for Molecular Recognition and Technological Development (IDM), Polytechnic University of Valencia, University of Valencia, 46022 Valencia, Spain; 6Department of Medicine, University of Valencia, 46010 Valencia, Spain; 7Department of Physiology, University of Valencia, 46010 Valencia, Spain

**Keywords:** HER2-positive, metastatic, clinical resistance

## Abstract

**Simple Summary:**

HER2-positive metastatic breast cancer remains a nearly incurable disease. In this sense, new treatments have been developed in recent years. On the one hand, drug conjugates have been reformulated and, on the other hand, the combination of anti-HER2 therapies with new drugs has been also tested. CDK4/6 inhibitors, tyrosine kinase inhibitors, and immunotherapy treatments have also been evaluated. Despite these advances, it is still urgent to continue deepening the biological knowledge of the disease and improving the therapeutic design of pharmacologic drugs in order to select the best available option for each patient.

**Abstract:**

HER2-positive breast cancer accounts for 15–20% of all breast cancer cases. This subtype is characterized by an aggressive behavior and poor prognosis. Anti-HER2 therapies have considerably improved the natural course of the disease. Despite this, relapse still occurs in around 20% of patients due to primary or acquired treatment resistance, and metastasis remains an incurable disease. This article reviews the main mechanisms underlying resistance to anti-HER2 treatments, focusing on newer HER2-targeted therapies. The progress in anti-HER2 drugs includes the development of novel antibody–drug conjugates with improvements in the conjugation process and novel linkers and payloads. Moreover, trastuzumab deruxtecan has enhanced the efficacy of trastuzumab emtansine, and the new drug trastuzumab duocarmazine is currently undergoing clinical trials to assess its effect. The combination of anti-HER2 agents with other drugs is also being evaluated. The addition of immunotherapy checkpoint inhibitors shows some benefit in a subset of patients, indicating the need for useful biomarkers to properly stratify patients. Besides, CDK4/6 and tyrosine kinase inhibitors are also included in the design of new treatment strategies. Lapitinib, neratinib and tucatinib have been approved for HER2-positive metastasis patients, however clinical trials are currently ongoing to optimize combined strategies, to reduce toxicity, and to better define the useful setting. Clinical research should be strengthened along with the discovery and validation of new biomarkers, as well as a deeper understanding of drug resistance and action mechanisms.

## 1. Introduction

Breast cancer (BC) is the most common cancer and the leading cause of cancer-related death among women worldwide. Based on the latest data published by GLOBOCAN, BC is the most prevalent cancer, with 2,261,419 cases diagnosed, constituting 11.7% of all global cancer cases in 2020 [1]. By the year 2023, experts project a total of 297,790 new cases of female BC. Within the spectrum of BC subtypes, HR+/HER2− subtype stands as the most prevalent, boasting an age-adjusted frequency of 87.2 new occurrences for every 100,000 women. This rate, calculated from cases spanning 2016 to 2020, surpasses other variants notably. To provide context, this rate is more than sixfold greater than the incidence of HR−/HER2− BC, which stands at 13.2 cases, as well as HR+/HER2+ BC, which stands at 12.6 cases. Even in comparison to HR−/HER2+ BC, with a rate of 5.1 cases, the HR+/HER2− subtype exhibits a remarkable more than seventeenfold higher frequency [2].

The human epidermal receptor type 2 (HER2) protein is a member of the ErbB/HER family of transmembrane tyrosine kinases, which also includes EGFR, HER3, and HER4, and is overexpressed in approximately 15% of all BCs. This characteristic confers aggressive behavior and a poor prognosis [3]. However, with the emergence of HER2-targeted therapies over the last few decades, patients’ prognoses have improved. Concretely, trastuzumab was the first HER2-targeting monoclonal antibody to be developed, and this advancement represents a significant achievement in the management of solid tumors.

The standard-of-care for HER2-positive BC involves a combination of targeted therapies and conventional treatments. Typically, the treatment approach includes anti-HER2 targeted therapies, chemotherapy, and hormonal therapy, if applicable [4]. Recently, the CLEOPATRA trial established the gold standard in the first-line setting with pertuzumab–trastuzumab–docetaxel, regardless of HR status [5].

Despite these new drugs, HER2-positive metastatic BC remains considered almost an incurable disease. Drug resistance (primary or acquired) is responsible for most treatment failures. Therefore, defining the best strategies to revert or delay anti-HER2 therapies resistance is crucial for improving clinical management.

In this review, we summarize current mechanisms of resistance and potential strategies to overcome them with novel therapies (Figure 1).

## 2. Mechanisms of Resistance and Response Heterogeneity to Anti-HER2 Therapies

### 2.1. PI3K/AKT/mTOR Pathway

The PI3K/AKT/mTOR pathway is considered an important cellular signaling mechanism involved in crucial cellular processes such as proliferation, survival, and metabolism [6]. Hyperactivation of this signaling pathway is associated with the development and progression of BC, including the HER2-positive subtype. This pathway is activated by the binding of extracellular ligands to specific receptors, leading to the phosphorylation of PIP2 and consequent formation of PIP3, which in turn recruits AKT and PDK1 to the plasma membrane. Finally, AKT phosphorylates target proteins to simulate cell survival, proliferation, and growth [7].

In BC, alterations in the PI3K/AKT/mTOR signaling pathway are common, including functional mutations in *PIK3CA* and loss of PTEN. This activation can occur through mutations or genetic alterations, resulting in increased cell survival and resistance to apoptosis. This hyperactivation leads to resistance to HER2-targeting therapies like trastuzumab. Consequently, with this biological rationale, the inhibition of the PI3K/AKT/mTOR pathway has been proposed as a therapeutic strategy to overcome resistance to anti-HER2 BC treatments. PI3K/AKT/mTOR pathway inhibition could improve the efficacy of anti-HER2 treatments as well as the clinical outcomes of HER2-positive BC patients. Further, the combination of PI3K inhibitor and trastuzumab has shown promising results in preclinical studies [8]. Moreover, everolimus, an mTOR inhibitor, is the first non-HER2-targeted therapy used to address trastuzumab resistance. Clinical trials have shown that the combination of everolimus, trastuzumab, and paclitaxel can improve progression-free survival (PFS) in patients with hormone receptor (HR)-negative and HER2-positive advanced BC [9]. Additionally, clinical trials of specific second-generation PI3K inhibitors are ongoing. The selective pan-Akt inhibitor ipatasertib (GDC-0068) is being tested in combination with pertuzumab and trastuzumab in BC patients with advanced HER2+ *PI3KCA* mutation [NCT04253561].

The pan-PI3K/mTOR inhibitor apitolisib (GDC-0980) has been investigated in conjunction with trastuzumab or trastuzumab emtansine (T-DM1) using cell lines and xenograft models. The combination with T-DM1 has exhibited superior efficacy and a more pronounced antitumor effect in both models [10]. Alpelisib (BYL719) is a highly potent and selective inhibitor of PI3Kα, with an IC50 of 5 nM in cell-free assays, and it exerts minimal impact on PI3Kβ/γ/δ. This compound is currently undergoing phase II clinical trials. The synergistic effect between alpelisib and anti-HER2 therapies has been demonstrated to enhance antitumor efficacy compared to monotherapy. This enhancement was observed both in vitro and in xenograft models [11].

### 2.2. p95HER2

Several studies have identified p95 as a potential mechanism of resistance to anti-HER2 therapy in BC. p95 is a truncated form of the HER2 receptor that lacks the extracellular domain and is constitutively active, leading to downstream signaling through the PI3K/AKT/mTOR pathway. This truncated form of HER2 is not recognized by trastuzumab, and it has been associated with resistance to this drug in preclinical models [12,13]. Furthermore, p95 expression has been found to be increased in HER2-positive BC patients who have relapsed after trastuzumab treatment [14]. All these findings suggest that p95 could be a potential biomarker for trastuzumab resistance in HER2-positive BC.

### 2.3. Insulin-Like Growth Factor I Receptor

Insulin-like growth factor I receptor (IGF-IR) is a tyrosine kinase receptor that regulates cell survival, proliferation, and transformation. Its activation has been demonstrated to promote HER2-targeted therapy resistance through several mechanisms, including modulation of the tumor microenvironment and activation of downstream signaling pathways [15,16].

Preclinical models demonstrate the efficacy of IGF-IR signaling inhibition in enhancing the efficacy of HER2-targeted therapies [17,18]. Ongoing clinical trials are investigating the combination of IGF-IR inhibitors with HER2-targeted therapies for BC treatment [19].

### 2.4. Tumor Heterogeneity

Intratumoral heterogeneity of HER2 expression has been described in 16–36% of HER2-positive BC patients and it is defined as the presence of varying degrees of HER2 overexpression in different areas of the same tumor [20]. Systemic treatment creates a dynamic molecular pressure that is involved in biological heterogeneity and may explain the adaptation in response to treatment exposure, leading to drug resistance. According to the literature, HER2 expression can oscillate from primary disease to metastasis by 9–60%, HER2 status being a clear example of cancer heterogeneity [21]. Sequential accumulation of genomic cell alterations induces oncogenic transformation that can affect biological functions such as proliferation, survival, migration, and invasion.

According to immunohistochemical (IHC) evaluation, HER2 may not be expressed homogeneously among all cancer cells in the same specimen. Its expression is defined as intense (3+) when complete circumferential membrane staining in more than 30% of all tumoral cells is present. However, in this case, a proportion of cells do not express HER2 on the cell membrane [22]. For this reason, heterogeneity plays an important role in the treatment of these patients. Metzger et al. classified as heterogeneous tumors those which had at least one area of HER2 negativity or HER2 positivity on a FISH test in fewer than 50% of total cells [23]. They demonstrated the absence of pathological complete response after HER2 treatment in the neoadjuvant setting in patients with heterogeneous tumors. In metastatic settings, the same scenario was observed, where equivocal or low-level HER2 amplification was significantly correlated with poor response to trastuzumab and T-DM1 [22]. These data suggest the importance of heterogeneity in treatment response. Modi et al. showed promising clinical antitumor activity of trastuzumab deruxtecan (T-DXd) in patients considered to be HER2-low or heterogeneous, probably due to the high levels of payload delivered in HER2-expressing cancer cells when compared with other anti-HER2 therapies [24].

### 2.5. Cyclin D1/CDK4/CDK6

Cyclin D1/CDK4/CDK6 may also be a mechanism of resistance to anti-HER therapies in BC. Cyclin D1 (CCND1) is a protein whose function is to regulate the G1 to S phase transition in the cell cycle, and is usually overexpressed in BC. This overexpression has been associated with poor response to anti-HER2 therapies [25]. CCND1 overexpression can activate CDK4 and CDK6, leading to increased cell proliferation and decreased sensitivity to anti-HER2 drugs [26]. Additionally, some studies have shown that CDK4/6 inhibition can improve the response to anti-HER2 therapies. Preclinical studies have demonstrated that the combination of CDK4/6 inhibitors with anti-HER2 therapies results in increased tumor growth inhibition and enhanced cell death in BC cells [27].

Overall, CCND1/CDK4/CDK6 overexpression may be a mechanism of resistance to anti-HER2 therapies in BC, and targeting CDK4/6 has shown promising results in overcoming this resistance.

### 2.6. HER2 Activating Mutations without Copy Number Gain

Bose et al. found HER2-activating mutations without gene amplification in some BC patients who had developed resistance to HER2-targeted therapies [28]. This study identified numerous *HER2* mutations, including L755S, L755P, V777L, S310F, and D769H, which were associated with resistance to anti-HER2 therapies. Another study by Ma et al. showed that L755S mutation can lead to constitutive activation of HER2 signaling, conferring resistance to trastuzumab and lapatinib in preclinical models [29]. These studies suggest that HER2-activating mutations without copy number gain may be a mechanism of resistance to HER2-targeted therapy in BC patients. Therefore, targeting *HER2* mutations in addition to HER2 amplification could improve the efficacy of HER2-targeted therapies for BC.

### 2.7. Overexpression of HER1 and HER3

HER1 and HER3 overexpression can activate signaling pathways that promote survival and cell proliferation despite the presence of anti-HER2 therapies. Preclinical studies have shown that dual (HER2 and HER3) or triple inhibition (HER1, HER2, and HER3) of HER receptors could overcome resistance and improve treatment response [30].

### 2.8. MET Aberrations

The *MET* gene encodes the protein hepatocyte growth factor receptor (HGFR). Aberrations or genetic alterations in this gene can lead to hyperactivation of the MET signaling pathway, resulting in cell growth, survival, and migration of cancer cells in metastatic BC [31]. Patients with BC who have *MET* aberrations have poorer responses to anti-HER2 therapies, lower survival, and worse prognosis than those without these aberrations [32]. Therefore, detecting *MET* aberrations in patients with metastatic BC could help identify those who may benefit from specifically targeted therapies against MET, such as MET inhibitors, thus improving their prognosis and quality of life.

### 2.9. Src

*Src* is a proto-oncogene that encodes the non-receptor protein tyrosine kinase Src, which plays a critical role in several cellular processes and promotes cell proliferation and survival [33]. Src interacts with transmembrane receptor tyrosine kinases (RTKs) located on the cell membrane, including HER1 and HER2. Thus, Src activation may lead to primary or acquired trastuzumab resistance. Some studies suggest that Src activation alone is sufficient to confer trastuzumab resistance once HER2 and Src are interconnected, and inhibition of HER2 can trigger Src activation [34]. Upon activation, Src can activate other signaling pathways that promote cancer cell survival and proliferation. Src activation has been associated with lower survival in HER2-positive BC patients [35]. Although new drugs that specifically target Src, such as dasatinib and saracatinib, are being developed to overcome anti-HER2 treatment-resistance in BC, more prospective studies are needed to demonstrate their efficacy [36].

In summary, Src activation may be a mechanism of resistance to anti-HER2 treatments in BC, and new drugs targeting Src could be an effective strategy to overcome this resistance and improve patient prognosis.

## 3. Newest HER2-Targeted Therapies

### 3.1. Improve Anti-HER2 Drugs

#### 3.1.1. Novel Antibody-Drug Conjugates

Nowadays it is necessary to highlight antibody–drug conjugates (ADCs). In recent years, novel ADCs have been approved for the treatment of solid tumors [Figure 2]. This has been possible due to improvements in the conjugation process, novel linkers, and payloads.

ADCs are sophisticated new molecules that target cell-surface proteins. They are composed by three components: an antibody directed to a cell surface protein, a cytotoxic agent (warhead or payload), and a linker used to connect the cytotoxic agent to the antibody [37]. This pharmacological approach improves drug delivery to tumors, reducing systematic toxicity [38]. However, these drugs induce bystander killing, which occurs when the cytotoxic element crosses the cellular membrane and acts directly against neighboring cells regardless of their HER2 expression level [20], which amplifies the magnitude of the therapeutic effect.

To date, thirteen ADCs have been already approved by the Food and Drug Administration (FDA) and several hundred are being investigated in clinical or preclinical assays (Table 1). Regarding anti-HER2 ADCs, most of them have trastuzumab or its modified version as an antibody [37].

##### T-DM1

T-DM1 was the first ADC approved for the treatment of a solid malignancy and consists of trastuzumab covalently linked via a non-cleavable linker to the cytotoxic agent DM1 (emtansine, a potent microtubule inhibitor). T-DM1 can release DM1 into tumor cells while maintaining the antitumor activity of trastuzumab [20]. The efficacy of T-DM1 was demonstrated in 3 phase II trials in HER2-positive patients. In the EMILIA phase III trial, TDM-1 was compared with lapatinib plus capecitabine as a second-line treatment in HER2-positive BC patients previously treated with taxane and trastuzumab [39]. The trial demonstrated that T-DM1 significantly improved PFS (9.6 vs. 6.4 months), decreasing the chance of death or disease progression in 35% of patients (95% CI, 0.55–0.77; *p* < 0.001), and overall survival (OS) (30.9 vs. 25.1 months; Hazard ratio, 0.68; 95% CI, 0.55–0.85; *p* < 0.001). These differences were observed regardless of the site of metastatic disease and HR status. Besides, T-DM1 presented a lower toxicity profile and an improved quality of life. Another phase III trial, TH3RESA, demonstrated improved PFS and OS with T-DM1 in patients who had received at least two anti-HER2 lines of therapy, including trastuzumab and lapatinib, in comparison with the physician treatment choice [40].

In 2013, these results led to FDA approval of T-DM1 for second and third-line treatment in patients who previously received trastuzumab and taxane in a metastatic setting or who experienced disease relapse within 6 months after the completion of adjuvant treatment. Thereafter, in May 2019, T-DM1 became the first-ever ADC approved for the adjuvant treatment of a solid malignancy based on the results of the KATHERINE trial [41]. This was a phase III study involving 1486 patients with HER2-positive locally advanced BC who had residual disease after neoadjuvant treatment. Patients were randomized to receive 14 cycles of adjuvant T-DM1 or trastuzumab. In this setting, T-DM1 improved invasive DFS (Hazard ratio, 0.50; three-year invasive DFS, 88.3% vs. 77%), with a 40% reduction in the risk of distant recurrence after a median follow-up of 41 months [42].

The role of T-DM1 as a standard second-line treatment has recently been challenged by the results of the DESTINYBreast03 phase III trial [43].

##### T-DXd

T-DXd is an ADC of a humanized anti-HER2 monoclonal antibody linked to a topoisomerase I inhibitor payload through a tetrapeptide-based cleavable linker. It is uniquely designed and has a high drug-to-antibody ratio (DAR) of approximately eight, that remains stable and confers a potent cytotoxic activity [44].

The proficient activity of T-DXd after failure of T-DM1 may be attributable to its pharmaceutical properties. Key biochemical differences compared to T-DM1 include the increased DAR, the different mechanism of action of the payload, which inhibits topoisomerase I instead of microtubules, and on the other hand, the potential induction of a bystander killing effect [42]. Its efficacy has been shown in phase II and III trials. DESTINY-Breast01, a phase II single-group study, demonstrated a durable antitumor activity in patients with pretreated metastatic HER2-positive BC. In that study, 60.9% (95% [CI], 53.4 to 68.0) of the patients who received T-DXd had an overall response (complete or partial response), and the median PFS was 16.4 months (95% CI, 12.7 to not reached) [16]. These results led to the accelerated approval of T-DXd in patients with metastatic HER2-positive BC who had received at least two prior lines of treatment. After the impressive antitumor activity observed in the DESTINY-Breast01 trial, it was necessary to perform a head-to-head trial to compare T-DXd with T-DM1. The DESTINY-Breast03 trial was the first phase III trial to compare the efficacy of two ADCs. In this trial, 84 HER2 metastatic BC patients treated with at least two prior lines of anti-HER2 therapy, including T-DM1 (median: 6 previous treatments), were treated with T-DXd. In intention-to-treat analysis, objective response rate (ORR) was 60.9%, with a median duration of response of 14.8 months, median PFS of 16.4 months, and OS at 6 and 12 months was 93.9% and 86.2%, respectively [45]. Regarding adverse events, 13.6% of patients presented interstitial lung disease, which was fatal in approximately 2.2%. Common grade III adverse events were gastrointestinal and hematological. In 2019, based on these results, FDA granted an accelerated approval to T-DXd for patients with HER2 metastatic BC who have received two or more prior anti-HER2-based regimens in the metastatic setting (Table 2).

##### HER2-Low BC

HER2-low disease is a new entity that presents an IHC score of 1+ or 2+ with a negative in situ hybridization score. A randomized open-label DESTINY-breast04 phase III trial was performed to evaluate T-DXd in HER2-low patients with unresectable or metastatic BC who were previously treated with one or two prior lines of chemotherapy [24]. The patients received T-DXd (5.4 mg/kg every three weeks) vs. physician’s choice chemotherapy. The addition of T-DXd significantly improved PFS and OS [46].

##### Trastuzumab Duocarmazine

Trastuzumab duocarmazine (SYD985) is an ADC that contains trastuzumab linked to the DNA alkylating agent seco-duocarmycin-hydroxybenzamide-azaindole. Interestingly, SYD985 has been shown to overcome T-DM1 resistance in both cellular and animal models [47]. Several trials are testing its action on different malignancies and stages. Some examples of these trials are the TULIP phase III trial (NCT03262935) in advanced or metastatic BC settings, a phase I/Ib trial in HER2-low tumors (ORR ranging from 28% to 40%, depending on HR expression) (NCT04602117) [48], and a phase I trial in patients with solid tumors, in which SYD985 is administered in combination with niraparib (NCT04235101) [37].

#### 3.1.2. Combinations of Anti-HER2 Agents with Other Drugs

##### Immunotherapy in HER2-Positive Disease

Recently, immunotherapy has changed the treatment of solid tumors by inducing hyperactivation of the immune response against tumors, resulting in better tumor regression and prolonged survival. Preclinical evidence supports the combination of HER2-targeted therapies with immunotherapy due to the synergistic activation of CD8 T cells [49]. ADCs induce lymphoid infiltration in the tumor microenvironment, upregulating the immune checkpoint receptors CTLA-4, PD-1, and PD-L1, and improving the immune response [50]. Based on this biological rationale, clinical trials have tested this hypothesis. The KATE2 phase II trial compared the safety and efficacy of T-DM1 alone or in combination with atezolizumab in HER2-positive patients with locally advanced BC who received prior trastuzumab and taxane therapy. Regarding PFS, the results were not significantly different when considering all patients without stratification. However, in PD-L1 positive subgroup, the PFS (8.5 months vs. 4.1 months) and one-year OS rate (94.3% vs. 87.9%) of patients receiving T-DM1 plus atezolizumab were significantly better compared to those receiving T-DM1 alone [42]. Moreover, exploratory analysis of immune-related biomarkers in this study (PD-L1 gene expression, effector T-cell gene signature, CD8 expression by immunohistochemistry, and tumor-infiltrating lymphocytes) showed that these could be associated with better PFS [51]. These data suggest that further investigation of HER2-targeted agents plus immune checkpoint inhibitors in a more selected subgroup of patients is necessary. Other clinical trials testing anti-HER2 therapies in combination with immune checkpoint therapies in earlier lines of treatment are ongoing [NCT03199885, NCT04538742].

##### CDK4/6 Inhibitors

In preclinical models, CDK4/6 inhibitors combined with trastuzumab have synergistic antitumor activity and could resensitize resistant HER2-positive BC to anti-HER2 therapies [52]. In this setting, MonarcHER and PATRICIA are two phase II trials that reported good results.

The first randomized study, MonarcHER [53], enrolled 237 HER2-positive and HR-positive advanced BC patients who had been previously treated with at least two HER2-targeted therapies. Herein, the efficacy of abemaciclib plus trastuzumab with or without fulvestrant was compared to chemotherapy plus trastuzumab. The treatment regime that showed better results was the triple combination (8.3 months vs. 5.7 months; Hazard ratio, 0.67; *p* = 0.051), whereas no differences were observed between abemaciclib plus trastuzumab and chemotherapy arm. On the other hand, the randomized PATRICIA trial, with 71 patients with HER2-positive BC who had been treated with at least 2–4 prior lines of anti-HER2 therapies, was performed [54]. These patients received palbociclib plus trastuzumab with or without letrozole (if HR-positive). This study demonstrated that combination therapy only benefits patients with HR-positive disease. Additionally, PAM50 associated the luminal intrinsic subtype with better PFS (10.6 months vs. 4.2 months; adjusted Hazard ratio, 0.40; *p* = 0.003) vs. non-luminal subgroup. For this reason, the ongoing trial PATRICIA-II (NCT02448420) aims to demonstrate the benefit of the triple strategy in PAM50 luminal disease [55].

A genetic mutation in *TSC2*, a gene responsible for regulating cell growth and division, results in resistance to anti-HER2 therapy. Through experiments conducted in both in vitro and xenograft models, the combination of palbociclib and lapatinib effectively surmounts this resistance triggered by the *TSC2* mutation. This research not only reveals a new mechanism of resistance but also offers an approach to surmounting it [56].

#### 3.1.3. TKI

Another class of anti-HER2 therapies is tyrosine kinase inhibitors (TKI). These drugs inhibit cellular growth by interrupting transduction cascades. To date, the FDA has approved three TKIs for the treatment of HER2-positive metastatic BC (lapatinib, neratinib, and tucatinib); however, further knowledge about their applicability in the context of early HER2-positive BC is needed [57].

##### Lapatinib

Lapatinib was the first TKI approved by the FDA for the treatment of HER2-positive metastatic BC. This TKI inhibits HER2 and epidermal growth factor receptor type 1 (EGFR1) and exerts its anti-tumor effects by blocking the HER2 signaling by competing with the antitumor ATP [58]. With this mechanism of action, this therapy is advantageous in overcoming resistance.

A phase III study compared lapatinib plus capecitabine with capecitabine alone in HER2-positive metastatic BC patients previously treated with anthracycline and a taxane and refractory to trastuzumab. The trial demonstrated a better PFS in combinatorial treatment (8.4 months vs. 4.4 months; Hazard ratio 0.49; 95% CI 0.34–0.071; *p* < 0.001), although no statistical differences were observed in OS [59]. Moreover, the EGF1001515 phase III trial demonstrated a significant improvement in PFS, from 4.3 to 6.2 months, in patients who received lapatinib plus capecitabine compared to those who received capecitabine alone (Hazard ratio, 0.55; 95% CI, 0.40–0.74; *p* < 0.001). However, no statistically significant differences were observed in OS (75.0 weeks for the combination arm vs. 64.7 weeks for the monotherapy arm [Hazard ratio, 0.87; 95% CI, 0.71–1.08; *p* = 0.210]). Another phase III trial (EGF104900 Study) showed the benefit of using trastuzumab plus lapatinib vs. lapatinib alone in patients with metastatic disease who were previously treated and showed progression to trastuzumab [60]. The combination treatment showed benefits in terms of PFS (12 weeks vs. 8.1 weeks; *p* = 0.008) and OS (14 months vs. 9.5 months; Hazard ratio 0.74; 95% CI 0.57–0.97; *p* = 0.026) compared to the lapatinib monotherapy arm.

The promising results observed in the clinical trials may be due to some lapatinib characteristics such as its small size, which allows entry into the systemic nervous system as it can easily cross the blood–brain barrier, thus improving the progression and response rates in brain metastasis, even in previously untreated brain metastasis.

##### Neratinib

Neratinib is an irreversible second-generation of pan-HER inhibitor (HER1, HER2, and HER4) that was firstly approved by the FDA for the adjuvant therapy setting based on the positive results of the EXTENET trial [29]. The NALA phase III trial included patients who had previously received at least two prior lines of anti-HER2 therapy and were randomized to receive either capecitabine plus neratinib or capecitabine plus lapatinib. The neratinib arm had significantly better PFS, with a 2.2-month absolute difference compared with those who received lapatinib (Hazard ratio 0.76; 95% CI 0.63–0.93; *p* = 0.0059). Moreover, neratinib improved ORR (32.8% vs. 26.7%; *p* = 0.12) and median response duration (8.5 vs. 5.6 months). However, no differences in OS were observed between arms. Based on these results, the FDA approved the use of neratinib as third-line therapy in metastatic disease [61]. Nevertheless, its use is often limited due to the high incidence of diarrhea as a secondary effect. Therefore, the CONTROL trial is being performed to improve its tolerability by administering a correct antidiarrheal prophylaxis, and dose escalation is being tested in order to decrease the incidence and severity of this toxicity [62].

##### Tucatinib

Tucatinib is a new TKI that is highly selective for the kinase domain of HER2 with minimal inhibition of EGFR, which explains its favorable toxicity profile. Moreover, its small molecular size facilitates the crossing of the brain–blood barrier, thus allowing direct activity against central nervous system metastatic disease.

Preclinical models showed good efficacy as monotherapy and in combination with chemotherapy and trastuzumab [63]. Moreover, xenograft models have shown that the addition of trastuzumab to tucatinib improves global response [64]. Based on the phase III HER2CLIMB study, tucatinib was approved by the FDA in April of 2020 for patients with HER2-positive metastatic BC. In the HER2CLIMB study, patients previously treated with anti-HER2 therapies (trastuzumab, pertuzumab, and T-DM1) were randomized 2:1 to receive tucatinib or placebo in combination with capecitabine and trastuzumab. After a median follow-up of 14 months, the median PFS was 7.8 months in the tucatinib arm vs. 5.6 months in the placebo arm, with a 46% decrease in the risk of death or progression (Hazard ratio, 0.54; 95% CI, 0.42–0.71; *p* < 0.001). The two-year OS rates were 44.9% in the tucatinib arm and 26.6% in the placebo arm. Interestingly, this trial included patients with active and treated brain metastasis (approximately 45% in both arms). In these cases, if patients presented local progression, treatment could be continued after local therapy. It is important to highlight the toxicity profile of this drug. In the tucatinib arm, diarrhea, transaminitis and hand–foot syndrome of any grade were frequent. However, only a small number of patients presented grade III or higher adverse events [65]. Evidence demonstrates that triple treatment is a highly effective option in this setting. Nowadays, there is growing interest in the combination of tucatinib with ADCs and several phase II trials are assessing the efficacy and safety of these combinations [NCT04539938 and NCT05673928].

FGFR4 is a cell surface receptor and tyrosine kinase that interacts with fibroblast growth factors, playing a pivotal role in regulating various pathways such as cell proliferation, differentiation, and migration. Research involving patient-derived xenografts and organoids reveals a synergistic interplay between anti-FGFR4 and anti-HER2 therapies in situations of both inherent and acquired resistance within BC. These findings collectively unveil a mechanism of resistance against anti-HER2 treatments and propose a potential strategy to counter this resistance by inhibiting FGFR4 in challenging cases of HER2-positive BC [66].

## 4. Future Directions

### 4.1. Biomarker Testing

Molecular technologies have identified biomarkers that can be used for diagnosis, prognosis, and treatment response prediction. BC biomarkers are mostly macromolecules such as DNA, RNA, proteins, and also cells. Identifying these biomarkers may help to overcome the challenge of drug resistance in BC.

miRNAs can be found in body fluids and have high stability, making them new promising diagnostic biomarkers. The use of signatures of circulating microRNAs (miRNAs) as diagnostic and prognostic biomarkers have shown better results than individual miRNAs [67]. Besides, DNA mutations in *ESR1*, *PARP*, *BRCA*, androgen receptor (a steroid HR), and CDK4/6 have been identified as potential biomarkers and are undergoing clinical validation [68].

The mechanisms underlying trastuzumab resistance are currently being investigated. Specifically, mucin 4 (MUC4) is a potential marker of resistance that interferes with the binding of trastuzumab to the HER2 receptor by covering the epitope. In 2009, Pohlmann et al. demonstrated, that MUC4 protein was associated with less antibody-binding capacity and epitope-masking driving trastuzumab resistance in the HER2-positive cell line JIMT-1 (intrinsic resistance to trastuzumab) [69].

Further, HER2 levels and tumor heterogeneity are becoming increasingly important biomarkers for both early and advanced BC. In the CLEOPATRA trial, it was found that patients with low HER2 expression levels had a worse PFS compared to those with high expression levels by IHC (Hazard ratio, 0.83; 95% CI, 0.69–1.00; *p* = 0.0502). However, this finding did not predict a better response to the combination of trastuzumab/pertuzumab compared to the single blockade, which is consistent with the results of the MARIANNE and TH3RESA trials. Although HER2 expression levels have substantial prognostic value, they are not predictors of the benefits of T-DM1 treatment [20].

In brief, these studies identified several potential biomarkers that could be useful for identifying HER2-positive BC patients who may have a higher likelihood of resistance to anti-HER2 therapies and who could benefit from alternative or combined therapies.

### 4.2. Liquid Biopsy

Liquid biopsy allows the detection and analysis of tumor biomarkers in the plasma of BC patients and it is considered a non-invasive technique. Regarding drug resistance, liquid biopsy is a promising tool for identifying biomarkers of resistance and guiding treatment. Plasma from BC patients was used to detect *HER2* gene amplifications and mutations in plasma of BC patients, demonstrating that the detection of mutations in other HER2 signaling-related genes such as *PIK3CA*, may predict resistance to anti-HER2 therapy.

The PLASMA-MATCH study is a phase II clinical trial that evaluates by liquid biopsy the efficacy of multiple therapies in patients with advanced treatment-resistant cancer [70]. This study employed next-generation sequencing technology to analyze liquid biopsy samples from patients with advanced cancer and to detect specific genetic mutations in the tumor. These results allowed the selection of a specific targeted treatment for the detected mutation. In the context of BC, the PLASMA-MATCH study evaluated the use of liquid biopsy to identify specific genetic mutations that may cause resistance to anti-HER2 therapy in patients with advanced HER2-positive BC. These results suggest that the liquid biopsy approach may be practical in selecting specific targeted treatments for patients with advanced HER2-positive BC resistant to anti-HER2 therapy. Besides, the detection of circulating tumor cells (CTCs) and circulating tumor DNA (ctDNA) in liquid biopsy has been investigated as a way to assess anti-HER2 therapy response and explain resistance. In fact, the presence of CTCs and ctDNA before and after treatment is associated with a worse prognosis and increased likelihood of developing resistance to anti-HER2 therapy [71].

Therefore, liquid biopsy is a promising tool for detecting biomarkers of resistance to anti-HER2 treatments in BC and for guiding therapy in patients with resistance or a high risk of developing resistance. However, further studies are needed to determine the clinical efficacy and utility of liquid biopsy in clinical practice.

### 4.3. AXL, Another Resistance Mechanism

As previously described, despite the clinical benefits of anti-HER2 drugs, many BC patients still develop resistance. In this sense, AXL has been recently identified as a new resistance mechanism. AXL is a member of the TAM family of receptor tyrosine kinases (TYRO3, AXL, and MERTK) that is implicated in the regulation of cell proliferation, migration, survival, and immune response [72]. Several studies have demonstrated that AXL is upregulated in various cancer types, including BC. Particularly in HER2 BC, AXL upregulation has been associated with anti-HER2 drugs. Its activation can bypass the HER2 signaling pathway promoting cancer cell survival and proliferation [73]. In addition, AXL can activate other downstream signaling pathways, such as PI3K/AKT and MAPK/ERK, which are involved in cancer cell growth and survival [74]. Preclinical models corroborate that AXL inhibition can overcome resistance to HER2-targeted therapies [75,76,77]. AXL upregulation can be considered a mechanism of resistance to HER2-targeted therapies in BC. Its inhibition has shown promising results in preclinical studies, and is currently being evaluated in clinical trials. AXL-targeted therapies may provide a novel approach to overcome resistance to HER2-targeted therapies in BC.

### 4.4. Diverse Approaches in Anti-HER2 Therapy Exploration

Further research in the realm of anti-HER2 therapy is centered around diverse approaches, encompassing bispecific antibodies, CAR-T, and strategies aimed at regulating protein production or degradation.

ZW25, a bispecific antibody, was developed to bind to two distinct domains of the HER2 protein, ECD4 (trastuzumab domain) and ECD2 (pertuzumab domain). In a phase I study involving heavily treated HER2-positive metastatic BC patients, ZW25 demonstrated a 33% response rate while maintaining good tolerability and no treatment discontinuations due to adverse effects. A phase II trial is currently underway to investigate ZW25’s effectiveness when combined with palbociclib and fulvestrant, aiming to provide chemotherapy-free options for advanced HR-positive/HER2-positive BC cases [NCT04224272] [78].

PRS-343, a bispecific T-cell engager (BiTE), is designed to engage CD137+ T-cells, crucial for immune stimulation, with HER2-positive tumor cells. This interaction boosts localized immune activity while reducing peripheral toxicity. Currently in a phase I clinical trial, PRS-343 has shown promising results, achieving a disease control rate of 58% with no severe grade three or four adverse effects reported [79].

ZW49 presents an innovative approach in the field of biparatopic ADCs, merging bispecific antibody technology with ADCs. ZW25 is crucial for targeting, enhancing precise payload delivery to cancer cells. ZW49’s unique payload, N-acyl sulfonamide auristatin, has shown encouraging preclinical outcomes, prompting the initiation of a phase I clinical trial [80].

Recent investigations show potential for CAR-T therapy in HER2-positive metastatic BC, particularly when combined with PD-1 antibodies. Although CAR-T therapy has demonstrated promise in hematologic cancers, its effectiveness against solid tumors has been limited. Current clinical trials are exploring CAR-T treatment for HER2-positive BC cases with brain or leptomeningeal metastases [NCT03696030] [81].

Targeted protein degradation (TPD) offers a novel approach to address proteins previously considered “undruggable”. Techniques such as PROTACs and molecular glues utilize natural cellular mechanisms for protein degradation. PROTACs, especially suitable for oral administration and reduced resistance risk, can selectively degrade proteins. Notably, a trastuzumab-PROTAC conjugate effectively triggers HER2 protein degradation in HER2-positive BC cases. The emergence of molecular glue technology further broadens the scope of targeted protein degradation, including HER2 and HER family proteins, offering innovative solutions to complex diseases [82,83].

## 5. Conclusions

Anti-HER2 therapies had a dramatic impact on the treatment landscape of HER2-positive BC. Trastuzumab, pertuzumab, and T-DM1 are commonly used in clinical practice. The appearance of non-responding cases and metastasis highlights the importance of improving actual treatment strategies to improve patients’ survival and quality of life. ADCs have long been the most successful in this context. T-DM1 was the first approved ADC for second- and third-line treatments. Thereafter, a second ADC, T-DXd, was effective after T-DM1 failure, showing remarkable antitumor activity in patients who had received prior lines of treatment. To enhance stability and potency while reducing adverse effects, more ADCs are being developed through new linker and conjugation technologies, thus improving chemistry and DAR. Furthermore, ADCs showed an optimal combinatorial effect with complementary drugs such as CDK4/6 inhibitors or TKIs. The development of brain metastases is a frequent event in HER2-positive BC patients, being more common than in HER2-negative BC patients. In this sense, some small molecular size TKI have shown promising responses in that setting of patients, where it may delay or complement local therapy strategies and represents a possible treatment option. Additional clinical trials with specific patient cohorts and defined endpoints are necessary to improve prognosis.

In addition, clinical trials combining immune checkpoint inhibitors and ADCs have shown benefit in specific subsets of BC patients. These results reveal the need to identify new biomarkers to improve patient selection.

Since there is no standard of care for metastatic BC in third and later lines of treatment, new treatment strategies are still needed. This involves the evaluation of new combinations of existing drugs, and the determination of reliable and easy-to-determine biomarkers of response. Finally, it is essential to increase the knowledge of new resistance mechanisms.

## Figures and Tables

**Figure 1 cancers-15-04522-f001:**
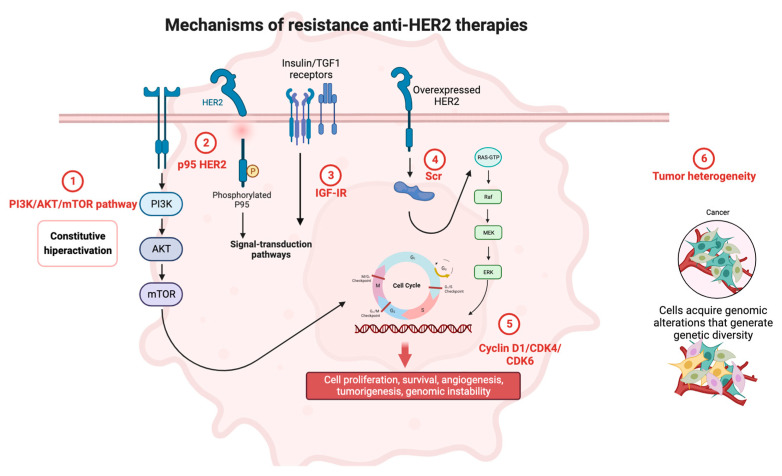
Mechanisms of resistance to anti-HER2 therapies. Created by Biorender.

**Figure 2 cancers-15-04522-f002:**
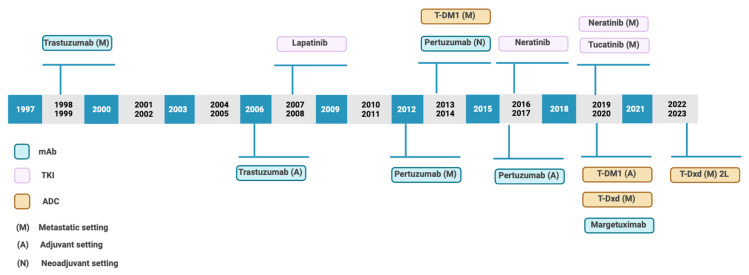
Timeline of approval anti-HER2 therapies. Based on Swain et al. [4].

**Table 1 cancers-15-04522-t001:** ADCs under development for HER2 BC.

ADC	Payload	Development Phase in Breast Cancer	NCT Identifier
Vic-trastuzumab duocarmazine	DUBA (DNA-alkylating agent)	III	TULIP (NCT03262935)
ARX788	AS269 (MT inhibitor)	II	ACE-Breast 03 (NCT04829604)
A166	Duo-5 (MT inhibitor)	I/II	KlusPharma (NCT03602079)
BDC-1001	TLR7/8 agonist	I/II	BBI-20201001 (NCT04278144)
ZW49	N-acyl sulfonamide auristatin (MT inhibitor)	I	ZWI-ZW49-101 (NCT03821233)
Disitamab-vedotin	MMAE (MT inhibitor)	I/II/III	NCT02881190, NCT03500380, NCT04400695)
Zanidatamab zovodotin	Auristatin based (MT inhibitor)	I	NCT03821233

MT: microtubules.

**Table 2 cancers-15-04522-t002:** Ongoing clinical trials with T-DXd including metastatic BC patients.

NCT Identifier	Trial Name	Phase	Experimental Arm	Control Arm	Setting	Primary Endpoint
NCT04538742	DESTINY-Breast 07	I/II	- T-DXd. - T-DXd plus durvalumab. - T-DXd plus pertuzumab. - T-DXd plus paclitaxel. - T-DXd plus durvalumab plus paclitaxel. - T-DXd plus Tucatinib.	-	≥2nd line	AEs
NCT04556773	DESTINY-Breast 08	I	- T-DXd plus capecitabine. - T-DXd plus durvalumab plus paclitaxel. - T-DXd plus capivasertib. - T-DXd plus anastrozole. - T-DXd plus fulvestrant tucatinib.	-	≥2nd line	AEs
NCT04494425	DESTINY-Breast 06	III	- T-DXd	Investigator’s choice standard of care chemotherapy (capecitabine, paclitaxel, nab-paclitaxel)	1st line	PFS
NCT04829604	ACE-Breast 03	II	- ARX788	-		ORR
NCT04784715	DESTINY-Breast 09	III	- T-DXd plus pertuzumab-matching placebo. - T-DXd plus pertuzumab.	Investigator’s choice standard of care (Taxane (paclitaxel or docetaxel), trastuzumab, and pertuzumab)	1st line	PFS
NCT05744375	TRASCENDER study	II	- T-DXd	-	2nd line (early relapse)	ORR
NCT04752059	T-DXd; DS-8201a	II	- T-DXd	-		ORR brain met
NCT04539938	HER2CLIMB-04	II	- Tucatinib plus T-DXd	-		ORR
NCT04739761	DESTINY-Breast 12	III	- T-DXd	-	No more than 2 lines	ORR

AEs: Adverse events. PFS: Progression-free survival. ORR: Objective response rate.

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
