# Peer review of "Clinical Impact of New Treatment Strategies for HER2-Positive Metastatic Breast Cancer Patients with Resistance to Classical Anti-HER Therapies"

_cancers, 2023, doi:10.3390/cancers15184522_

Round 1
Reviewer 1 Report
HER2-positive metastatic breast cancer remains incurable. In the past decades, new treatments have been developed in reversing the antiHER2 resistance such as the new development of ADC , TKI, et al. the combination of anti-HER2 therapies with new drugs has been also tested, Such as CDK4/6 inhibitors, tyrosine kinase inhibitors, and immunotherapy. It is still urgent to continue deepening the biological knowledge of the disease and improving the therapeutic design of pharmacologic drugs in order to improve prognosis and provide the best personal treatment for each patient.
This article provides a comprehensive review of the mechanism of antiHER2 resistance, the development of new antiHER2 drugs and new combinations, and the future research direction, which will give great helps for us to understand and explore new solutions for overcoming anti-her2 resistance. The paper is very well written, and well organized. However, there are still some problems, which must be solved before it is considered for publication.
Major revisions:
1. This paper is about the clinical impact of new treatment strategies for HER2-positive metastatic breast cancer patients with resistance to classical anti-HER therapies, what kind of drugs means classical anti-HER therapies. Does classical HER2 resistance mean Herceptin resistance or Herceptin and pertuzumab resistance? If it means trastuzumab resistant, the interaction between HER2 and other family members should be also involved in the part of mechanism.
2. There is a lack of elaboration on HER3 target therapy in the review, such as the research of HER3-DXD.
3. In terms of drug combination, the combination of anti-HER2 therapy and PAM pathway inhibitors also have some clinical evidence which needs to be described.
4. The content is not comprehensive enough. For example, in the future direction other than biomarker, liquid biopsy and AXL, there are some new progress in new treatment method and combinations. the literature needs to be updated. For future solutions, no more new and important solutions are proposed.
5. There have been a number of similar reviews, most of which are repetitive and need to be updated by the authors to distinguish them from other reviews.(CancerSiddharth Kunte, Jame Abraham, Alberto J Montero. Novel HER2-targeted therapies for HER2-positive metastatic breast cancer.2020 Oct 1;126(19):4278-4288. doi: 10.1002/cncr.33102. Epub 2020 Jul 28. )
Minor revisions:
1. In line 252 -254: The phase III trial known as DESTINY Breast03 show cased a substantial and clinically meaningful advantage of T-DXd over T-DM1 as a second-line treatment, as described below, questioning the use of T-DM1.
The mention of DS8201 after the introduction of DS8201 maybe more logical, for reference only.
2. In the paragraph of TDXD, the author mentioned low HER2 expression. Almost all patients with low HER2 expression have not received anti-HER2 treatment, is it appropriate to include it in this article.
3. In the mechanism of resistance, the influence of HER2 heterogeneity on the efficacy was discussed. Whether the shift of HER2 from positive to negative has some effect on treatment resistance and how about the corresponding treatment recommendation?
Author Response
Dear reviewer,
Your comments to improve the manuscript are appreciated. We have modified the text accordingly following your recommendations. The answers to the questions can be found attached.

Reviewer 2 Report
In this review article, Tapia M et al have summarized information about the various resistance mechanisms that contribute to treatment failure in HER2 positive breast cancer. In addition, they have also provided information about the recent advances made in developing new therapeutic strategies to treat this debilitating disease. Overall, the review is quite extensive and includes the latest information pertaining to treatment modalities available for managing HER2 positive breast cancer. Suggestions for the authors are listed below:
1. The Introduction/ background section of the review is extremely short and it is recommended that the authors provide information about the epidemiology of the disease with the recent statistics.
2. Further, the authors should also provide a rationale as to why did they focus only on HER2 positive breast cancer.
3. Also, the authors also need to describe the standard-of-care that is used for management of the disease in the introduction section
4. It is unclear from the text whether the treatment options described are only applicable in resistance setting or are part of the second-line and/ or third-line treatment options.
The quality of English needs improvement.
Author Response

(The authors gave the same response as above.)

Round 2
Reviewer 2 Report
The authors have addressed the comments and suggestions. The manuscript looks good.
Minor checks required